# The time-dependent changes in a mouse model of traumatic brain injury with motor dysfunction

Dohee Kim[1], Jinsu Hwang[1], Jin Yoo[2], Jiyun Choi[1], Mahesh Ramalingam[1], Seongryul Kim[1], Hyong-Ho Cho[3], Byeong C. Kim[4], Han-Seong Jeong[1]*, Sujeong Jang[1]*

1 Department of Physiology, Chonnam National University Medical School, Gwangju, Jeollanamdo, Republic of Korea, 2 Department of Physical Education, Chonnam National University, Gwangju, Republic of Korea, 3 Department of Otolaryngology-Head and Neck Surgery, Chonnam National University Hospital, Chonnam National University Medical School, Gwangju, Republic of Korea, 4 Department of Neurology, Chonnam National University Hospital, Chonnam National University Medical School, Gwangju, Republic of Korea

* jhsjeong@hanmail.net (HSJ); sujeong.jjang@gmail.com, sujeongjang@jnu.ac.kr (SJ)

**Data Availability Statement:** All relevant data are within the paper and its Supporting Information files.

## Abstract

Traumatic brain injury (TBI) results from sudden accidents, leading to brain damage, subsequent organ dysfunction, and potentially death. Despite extensive studies on rodent TBI models, there is still high variability in terms of target points, and this results in significantly different symptoms between models. In this study, we established a more concise and effective TBI mouse model, which included locomotor dysfunctions with increased apoptosis, based on the controlled cortical impact method. Behavioral tests, such as elevated body swing, rotarod, and cylinder tests were performed to assess the validity of our model. To investigate the underlying mechanisms of injury, we analyzed the expression of proteins associated with immune response and the apoptosis signaling pathway via western blotting analysis and immunohistochemistry. Upon TBI induction, the mouse subjects showed motor dysfunctions and asymmetric behavioral assessment. The expression of Bax gradually increased over time and reached its maximum 3 days post-surgery, and then declined. The expression of Mcl-1 showed a similar trend to Bax. Furthermore, the expression of caspase-3, ROCK1, and p53 were highly elevated by 3 days post-surgery and then declined by 7 days post-surgery. Importantly, immunohistochemistry revealed an immediate increase in the level of Bcl-2 at the lesion site upon TBI induction. Also, we found that the expression of neuronal markers, such as NeuN and MAP2, decreased after the surgery. Interestingly, the increase in NFH level was in line with the symptoms of TBI in humans. Collectively, our study demonstrated that the established TBI model induces motor dysfunction, hemorrhaging, infarctions, and apoptosis, closely resembling TBI in humans. Therefore, we predict that our model may be useful for developing effective treatment option for TBI.

**Funding:** This research was supported by grants from the National Research Foundation of Korea (Grant Numbers NRF-2021R1I1A3060435 and NRF-2020R1F1A1076616); a Grant from the Chonnam National University Hospital Biomedical Research Institute (BCRI23041); a Grant from the Korea Institute for Advancement of Technology (KIAT, grant number P0020818) funded by the Korean Government (MOTIE); and a Grant from the Jeollanam-do Science and Technology R&D Project (Development of stem cell-derived new drug) funded by the Jeollanam-do, Korea. The funders had no role in study design, data collection and analysis, decision to publish, or preparation of the manuscript.

**Competing interests:** The authors have declared that no competing interests exist.

## Introduction

Traumatic brain injury (TBI) occurs when the brain becomes damaged by external forces, such as falls, car accidents, struck objects, violence, and wars [1–4]. According to the World Health Organization (WHO), there are approximately 1,300 cases of TBI per 100,000 people in North America and 170,000 people in the Republic of Korea annually [5]. TBI is currently the third leading cause of death globally, as it can lead to organ dysfunction depending on the severity of the injury [6–8]. Many people suffer during a short or long time for TBI, which is affecting behavior, cognition, and motor skills [9–11]. TBI is categorized into two types—open-head injury and non-penetrating injury. The open-head injuries penetrate the skull and cause serious problems that eventually lead to death when no treatment option is available [12]. The non-penetrating injuries cause mild TBI, and their typical symptoms include anxiety, depression, irritability, and headache [13, 14].

In TBI, irreversible primary injury is produced by mechanical forces and subsequently followed by the secondary injuries [15–17]. Sustained damage to neurons and glial cells results in cell death, ultimately leading to dysfunction in related organs, such as the brain and spinal cord within the central nervous system (CNS) [18]. Therapeutic strategies for treating secondary injuries include pharmacological interventions with neuroprotective and anti-inflammatory agents, as well as the control of cerebral blood flow and intracranial pressure [5, 15, 19–21]. However, these strategies do not always lead to complete recovery from TBI [22–24]. To validate the effective treatment options for TBI, a more comprehensive and accurate modeling in animals must be performed before the pre-clinical trials. Despite extensive studies on rodent TBI models, there is still high variability in terms of target points, and this results in significantly different symptoms between the models. As each symptom is associated with a unique molecular mechanism, such as differences between the models make the elucidation of effective treatment options difficult.

The Bcl-2 family consists of proteins secreted in mitochondria. The changes in the intracellular levels of these proteins induce the activation of caspases that promote apoptosis, become activated [25, 26]. Particularly for neurons, caspase activation in associated with cell death, in which the use of a caspase inhibitor has been demonstrated to mitigate traumatic apoptosis and neurological deficits [27].

In this study, we established an accurate mouse model of TBI with motor dysfunction based on the controlled cortical impact (CCI) method, which is one of the focal-impact TBI modelling methods. This method targets the region between bregma and lambda in the skull [28–30] and is considered more delicate than the weight drop method as it can control factors like speed, depth, and dwell time. Our established model was validated by investigating the symptomatic differences between the control, sham, and TBI-induced mice groups. Specifically, we investigated the changes affecting neurons and glial cells at 3 hours, 1, 3, 7, and 14 days after the TBI-inducing surgery.

## Materials and methods

### Animals

All experiments were approved (approval No. CNU IACIC-H-2023-11) and followed the animal ethics guidelines of the Institutional Animal Care and Use Committee of Chonnam National University Medical School, Jeollanamdo, Republic of Korea. For surgical procedure, we selected a C57BL/6J mouse strain, which shows better coordination than C57BL/6N. In addition, many researchers, who studied about central nervous system (such as brain and spinal cord) injuries, used this mouse strain in the TBI [31–34]. Male C57BL/6J mice (20–30 g,

12–14 weeks old) were purchased from Damul Science (Daejeon, Republic of Korea) and randomly assigned to seven groups using a random table method: (1) non-surgery and sedentary group (Control or Con, n = 20); (2) Sham-surgery and sedentary group (Sham, n = 20); (3) 3-h post-TBI and sedentary group (TBI-3h, n = 20); (4) 1-d post-TBI and sedentary group (TBI-1d, n = 20); (5) 3-d post-TBI and sedentary group (TBI-3d, n = 20); (6) 7-d post-TBI and sedentary group (TBI-7d, n = 20); (7) 14-d post-TBI and sedentary group (TBI-14d, n = 20). All mice were housed in the animal room with access to food and water at 22–24°C under a 12-hour light/dark cycle.

## Surgical procedure

After one week of adaptation, a surgical procedure was performed on the sham and TBI groups modified from previous studies [28, 30–37]. In brief, the subject was anesthetized with pentobarbital sodium (50 mg/kg; JW Pharmaceuticals, Seoul, Republic of Korea) and stabilized with 1–2% of isoflurane (ISOTROY, Troikaa Pharmaceuticals, Gujarat, India) during the surgical process. The TBI group received surgery of 1.00 mm in depth, 4.00 m/s in speed, and 300 ms in dwell time. The diameter of the hammer used for TBI inductions was 2.0 mm and the target point was at– 0.1 mm from bregma and + 0.8 mm from midline or sagittal suture. After fixing the brain, an impactor was directly targeted to the point (68099 II precision impactor device, RWD Life Science, San Diego, CA, USA). The control did not receive any surgical process, where the sham group received surgery of 0.00 mm in depth (Fig 1).

## Behavioral tests

**Elevated body swing test.** The behavioral tests were performed 1 day before the surgery and 0, 1, 3, 5, 7, and 14 days after the surgery. For the elevated body swing and cylinder tests conducted before the surgery, the subjects were divided into three groups to prevent bias toward left or right in the same group. For the rotarod test, the mice were trained thrice per day for 3 days before surgery.

An elevated body swing test has a long history and is a very simple asymmetric behavioral test [38, 39]. In brief, a mouse was placed in an empty cage and allowed to adjust to a neutral position with four paws on the ground. Next, the mouse was raised by the tail about 1 inch from the ground. The direction of the mouse subject's body swing in the air, >10° bending along the left or right axis, was recorded. The test was performed 20 times for each subject and the score was calculated by dividing the number of left swings by the total number of swings.

**Rotarod test.** The rotarod test is suitable for evaluating the motor coordination of mice [40, 41]. In brief, the mouse subject was placed on the rotating drum that was gradually accelerated from 4 to 40 rpm over 5 minutes. Whenever the mouse fell from the drum, the time and speed were recorded. The test was repeated 3 times and the results were average. The subjects were given sufficient rest between the test.

**Cylinder test.** The cylinder test was performed to assess the subject's forelimb asymmetry. The subject was placed in a transparent cylinder with a diameter of 9 cm and a height of 15 cm [42]. Briefly, we counted the number of times the subject's forelimb made with the cylinder wall. Contacts were only recorded when the forelimb was fully extended and touched the wall 20 times. Subsequently, the percentage of left forelimbs usage was calculated.

## Triphenyltetrazolium chloride (TTC) staining

After TBI-inducing surgery, the brain of the subjects had primary and secondary infarction regions. To determine the size of these regions, we stained the sections of the brains with TTC (Sigma Aldrich, Burlington, MA, USA, Cat No. T8877). In brief, the subject was anesthetized

**A** Part 1. Surgery and Behavioral test

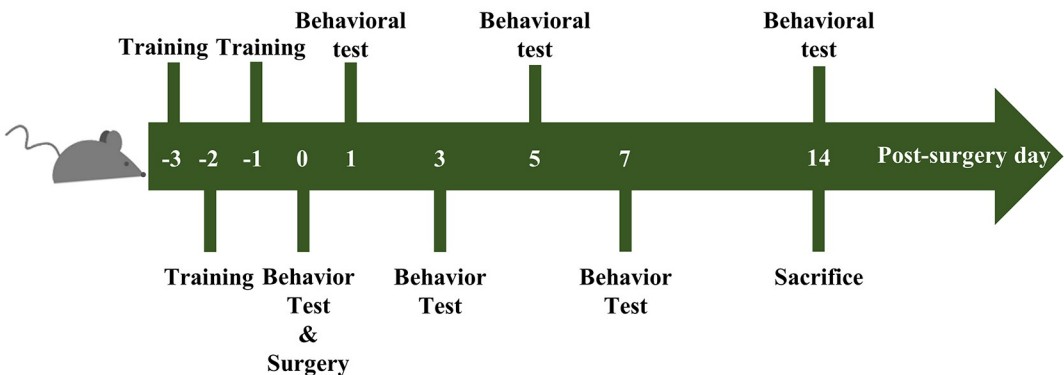

**B** Part 2. TTC, IHC, Western blotting

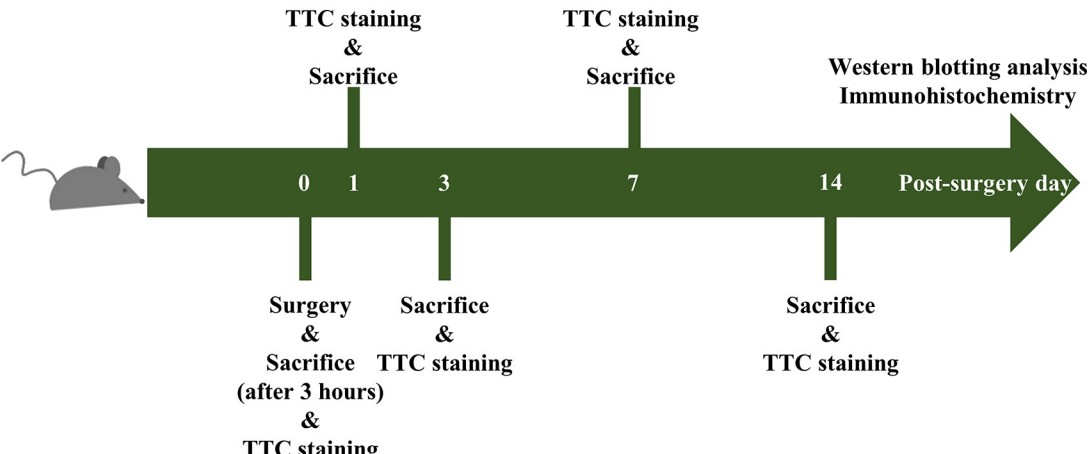

**Fig 1. Schematic illustration of the study.** (A) Part 1: the mice were divided into three groups and trained for 1, 2, and 3 days. Next, the surgical operation took place on Day 1, and the behaviors of the subjects were observed on Days 1, 3, 5, 7, and 14. (B) Part 2: the surgical operation took place on Day 0. The subjects were euthanized 3 hours, 1, 3, 5, 7, or 14 days post-surgery. At each time point, the subjects' brain tissues were harvested and the samples were stained with TTC for whole-brain imaging.

with sodium pentobarbital, euthanized, and its brain was harvested. The brain was then cut into three slices of 2 mm thickness, stained with 2% TTC in PBS, and incubated in an incubator at a 37°C for 30 minutes as previously described [43]. Subsequently, the sections were fixed with 4% paraformaldehyde (PFA, Biosesang, Seongnam, Republic of Korea, Cat No. pc2031-100-00) for 1 to 24 hours and imaged.

## Western blotting analysis

To investigate the protein level in each group, we carried out a western blotting analysis following our previous study [44]. After euthanasia, the brains were harvested, and protein were extracted using RIPA Lysis and Extraction Buffer (RIPA buffer, Thermo fisher, Waltham, MA, USA, Cat No. 89900). Pierce BCA protein assay kits (Thermo fisher, Cat No. 23225) was used

**Table 1. List of antibodies [44–46].**

| Name | Company | Cat No. | Dilution | Application |
|---|---|---|---|---|
| Bax | Santa cruz | SC-493 | 1:3000, 1:400 | WB, IHC |
| Bcl-2 | Santa cruz | SC-492 | 1:500, 1:200 | WB, IHC |
| NeuN | Cell signaling | CST#24307 | 1:200 | IHC |
| MAP2 | Cell signaling | CST#8707 | 1:400 | IHC |
| NFH | Cell signaling | CST#2836 | 1:400 | IHC |
| GFAP | Cell signaling | CST#3670 | 1:1000 | IHC |
| Iba1 | Cell signaling | CST#17198 | 1:500 | IHC |
| Mcl-1 | Cell signaling | CST#94296 | 1:1000 | WB |
| Caspase 3 | Cell signaling | CST#9665 | 1:3000 | WB |
| Cleaved-caspase 3 | Cell signaling | CST#9664 | 1:1000 | WB |
| ROCK 1 | Cell signaling | CST#4035 | 1:1000 | WB |
| p53 | Santa cruz | SC-126 | 1:500 | WB |
| β-actin | Cell signaling | CST#5125 | 1:3000 | WB |
| Biotinylated horse anti-Mouse IgG | vector | PK-6102 | Same the primary antibodies concentration | IHC |
| Biotinylated goat anti-Rabbit IgG | vector | PK-6010 | Same the primary antibodies concentration | IHC |
| Anti-mouse IgG | Cell signaling | CST#7076 | 1:1000 | WB |
| Anti-rabbit IgG | Cell signaling | CST#7074 | 1:1000 | WB |

to determine the protein concentration of each sample, with 20 μg/mL of proteins being used in this study. Next, the extracted proteins were separated on 11% SDS-polyacrylamide gels and then transferred onto polyvinylidene difluoride membrane (Millipore, Bradford, MA, USA, Cat No. IPVH00010). The membranes were incubated with primary antibodies at 4°C for overnight and then treated with a secondary antibody. To detect the proteins, the membranes were treated with an enhanced chemiluminescence reagent (ECL, Millipore, Billerica, MA, USA, Cat No. WBLUR0500) under luminescent image analyzer (LAS 4000, GE Healthcare, Little Chalfont, UK). The primary and secondary antibodies used in this study are listed in Table 1.

## Preparation of the brain

To investigate the changes in the brain upon TBI induction, the subjects were first euthanized by perfusion following our previous study [47]. In brief, the mouse was anesthetized with sodium pentobarbital, transcardially perfused with 4% PFA, and the brain was harvested. The brain was embedded in a mold with optimal cutting temperature compound (OCT, Leica Bio-system, Wetzlar, Germany, Cat No. 3801480) and cut into slices of 10 μm thickness using a cryostat (Leica CM1860, Leica Biosystems).

## Hematoxylin and eosin (H&E) staining

To investigate the lesion sites, H&E staining was performed following our previous study [48]. The tissue sections were stained with Harris hematoxylin solution (Dako, Glostrup, Denmark, Cat No. S3309) for 8 minutes, followed by washing with distilled water for 5 minutes. Subsequently, the sections were further washed with distilled water for 5 minutes, followed by counterstaining with eosin Y (Duksan, Seoul, Republic of Korea, Cat No. d1398) for 30 seconds. The sections were dehydrated and mounted for imaging analysis. Hematoxylin-stained cell nuclei displayed a purplish-blue color, while eosin-stained extracellular matrix and cytoplasm displayed a pink color.

### Diaminobenzidine (DAB) staining

To investigate the apoptosis-associated protein and neuronal marker levels in the brain of the TBI-induced subjects, DAB staining was performed as previously reported [47]. In brief, the tissue sections were incubated with primary antibodies against Bax, Bcl-2, NeuN, and MAP2, which are listed in Table 1. The sections were then incubated with a secondary antibody and reacted with an avidin-biotin complex (Vector, Cat No. PK-6100). Next, a DAB substrate kit (DAB, Enzo life sciences, Farmingdale, NY, USA, Cat No. ENZ-KIT159-0150) was treated to the sections. The nuclei were stained with hematoxylin as previously described.

### Statistical analysis

All statistical analyses were performed using GraphPad Prism 5 software (San Diego, CA, USA). For comparative analyses of the infarction regions and protein levels, one-way ANOVA was utilized. The significance of behavioral deficits results was determined by two-way ANOVA analysis. Each TBI group was compared against the Con or Sham group. A p-value of less than 0.05 was considered significant.

## Results

### Behavioral dysfunctions upon TBI induction

To validate the proper modeling of TBI with behavioral dysfunctions, we analysed the behaviors of TBI-induced mice at different time points after the surgery. As shown in Fig 2, we performed the rotarod (Fig 2A), elevated body swing (Fig 2B), and cylinder tests (Fig 2C) for up to 14 days after the surgery. The motor coordination of the subjects, as tested by the rotarod test, rapidly declined as early as 1-day post-surgery (Fig 2A). The asymmetry of the subjects, as assessed by the elevated body swing and cylinder tests, started around the same time (Fig 2B and 2C). While the motor coordination of the subjects seemed to recover slightly over time, their asymmetry was sustained for an extended duration and appeared to be severe.

### Increased brain infractions and secondary injuries

To assess the brain infarction and its changes over time, we investigated the hemorrhaging in the brain (Fig 3). The experimental design is depicted in Fig 1B. Following TBI induction, the subjects' brains were wounded, with hemorrhaging gradually increasing over time (Fig 3A). TTC staining was conducted to investigate the sized of the infarction regions and secondary

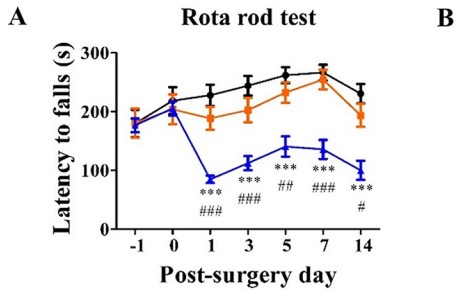
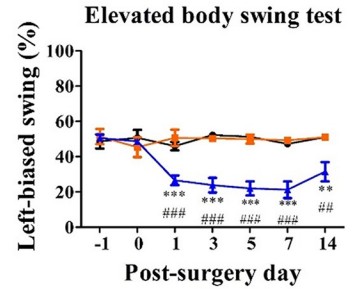
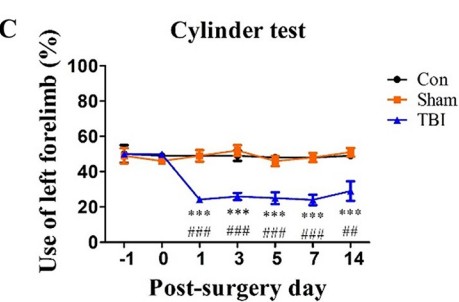

**Fig 2.** Evaluation of motor dysfunctions after TBI induction with rotarod test (A), elevated body swing test (B), and cylinder test (C). (A) The rotarod test indicated motor disturbances in TBI-induced subjects, while the elevated body swing and cylinder tests indicated their asymmetric actions. (B-C) The plotted results of each behavioral test. After TBI induction, motor dysfunctions were observed in the subjects. The rotarod test showed a rapid decrease in motor functions for the TBI group, which failed to recover over time. Moreover, the TBI group showed increasing severity in their asymmetric action over time. *$p < 0.05$, **$p < 0.01$, ***$p < 0.001$ compared to the Con group; #$p < 0.05$, ##$p < 0.01$, ###$p < 0.001$ compared to the Sham group (n = 20).

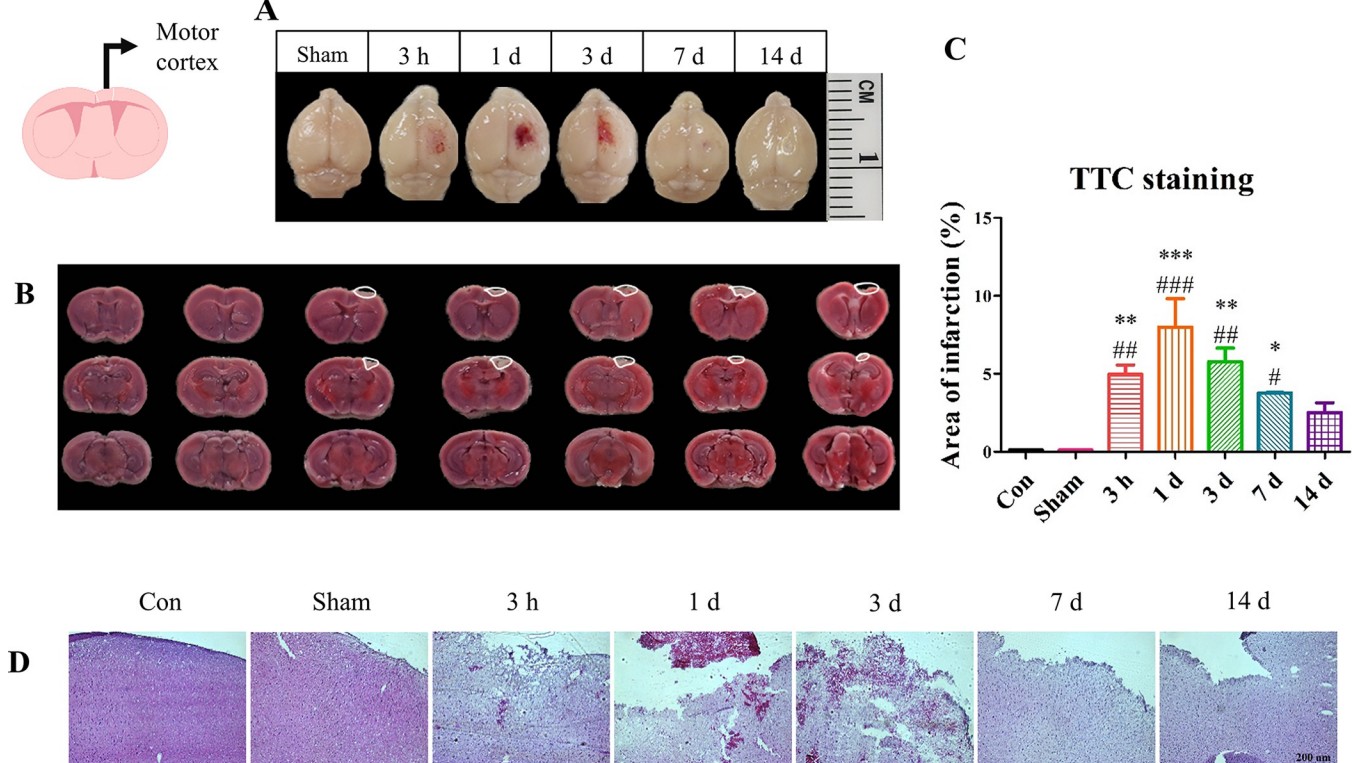

**Fig 3. The transient changes in the brain after TBI induction.** (A) Whole-brain imaging was carried out 3 hours, 1, 3, 7, and 14 days post-injury. In contrast to the Sham group, the TBI group showed signs of hemorrhaging, which reached its peak on Day 1 and gradually decreased by Day 3. (B) In the TTC-stained images, the white line indicates the hemorrhage lesions and infarction areas. The empty sites are the hemorrhage lesions, while the white color sites are the infarction areas. The infarction areas gradually increased until Day 1, slightly decreased by 3 days, and virtually disappeared by Day 14. (C) The quantified infarction areas for different time points post-surgery. (D) The H&E stained images of each group. After the surgery, the cytosol showed signs of collapsing and aggregation, which were recovered by Day 7. Each experiment included three biologically independent mice per group. Significance levels: $*p < 0.05$, $**p < 0.01$, $***p < 0.001$ compared to the Con group; $\#p < 0.05$, $\#\#p < 0.01$, $\#\#\#p < 0.001$ compared to the Sham group.

injuries (Fig 3B and 3C). The brains were each sectioned into three slices. The Con and Sham groups showed no sign of infarction (Fig 3B). On the other hand, the TBI group showed notable signs of infarction (Fig 3B), as well as secondary injuries in the surrounding regions. The infarction regions reached the maximum at 1-day post-surgery and decreased over time. By 14 days post-surgery, hemorrhaging seemed to have disappeared, suggesting that some level of spontaneous healing had occurred even after the induction of TBI. However, as shown in Fig 2, it was evident that such healing of hemorrhaging and infarctions did not lead to a complete recovery from behavioral disorders.

## TBI induction stimulates the apoptosis signaling pathways

Upon TBI induction, we investigated the changes in the levels of proteins associated with the apoptosis signaling pathway at different time points (Figs 4 and 5). The expression of Bax, a pro-apoptotic factor, was elevated at 3 days post-surgery and subsequently decreased by 3 days after that point (Fig 4C and 4F). The expression of Bcl-2, an anti-apoptotic factor, was slightly decreased after the surgery, but not a significant degree (Fig 4B and 4E). The Bax/Bcl-2 ratio gradually increased and reached the maximum at 3 days post-injury, and then decreased by 7 days post-surgery (Fig 4G). The expression of Mcl-1, an anti-apoptotic Bcl-2 family protein,

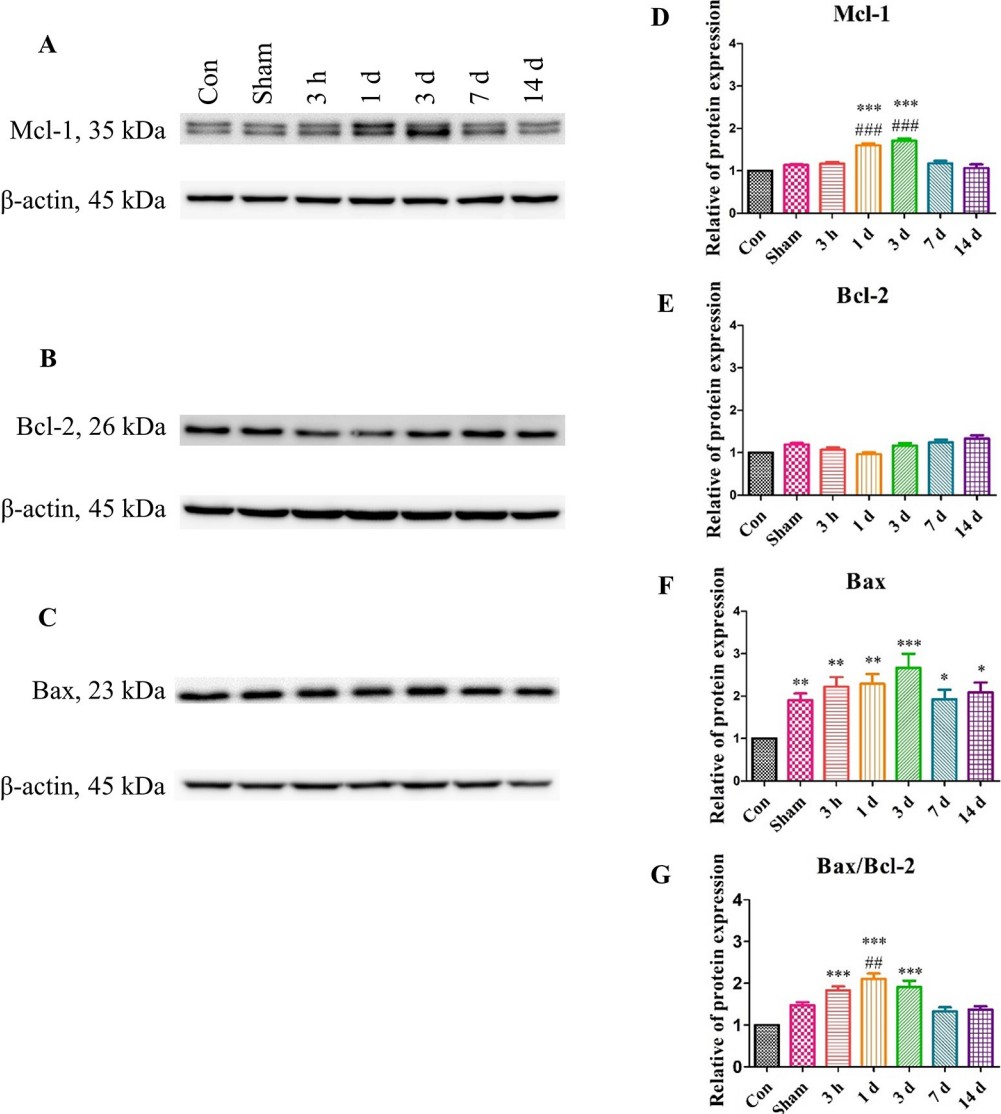

**Fig 4. The change in the Bcl-2 family proteins' levels in the brain after TBI induction.** (A-C) Western blotting analyses. (D-F) The quantified western blotting results. (G) The relative expresssion levels of Bax/Bcl-2 was calculated and quantified. All protein levels were normalized to that of β-actin and fold change was calculated relative to the Con groups. The western blotting analyses were carried out 3 hours, 1, 3, 7, and 14 days post-surgery. Each experiment included three biologically independent mice per group. Significance levels: $^*p < 0.05$, $^{**}p < 0.01$, $^{***}p < 0.001$ compared to the Con group; $^#p < 0.05$, $^{##}p < 0.01$, $^{###}p < 0.001$ compared to the Sham group.

also increased after the surgery similarly to the Bax/Bcl-2 ratio (Fig 4A and 4D). Mcl-1 is an anti-apoptotic member which interacts with Bcl-2 family and inhibits apoptosis. They showed two bands and it is same results from the previous studies [44, 46].

Furthermore, the Caspase3-to-cleaved Caspase3 ratio, ROCK1, and p53 proteins, downstream factors in apoptotic signaling, were elevated in the TBI group (Fig 5). This result was consistent with the aforementioned results to a certain extent. The western blotting analysis revealed that the induction of TBI stimulated the Bcl-2 family proteins and apoptosis signaling pathway. Furthermore, it appeared that the secondary injury could also lead to the stimulation of apoptosis signaling.

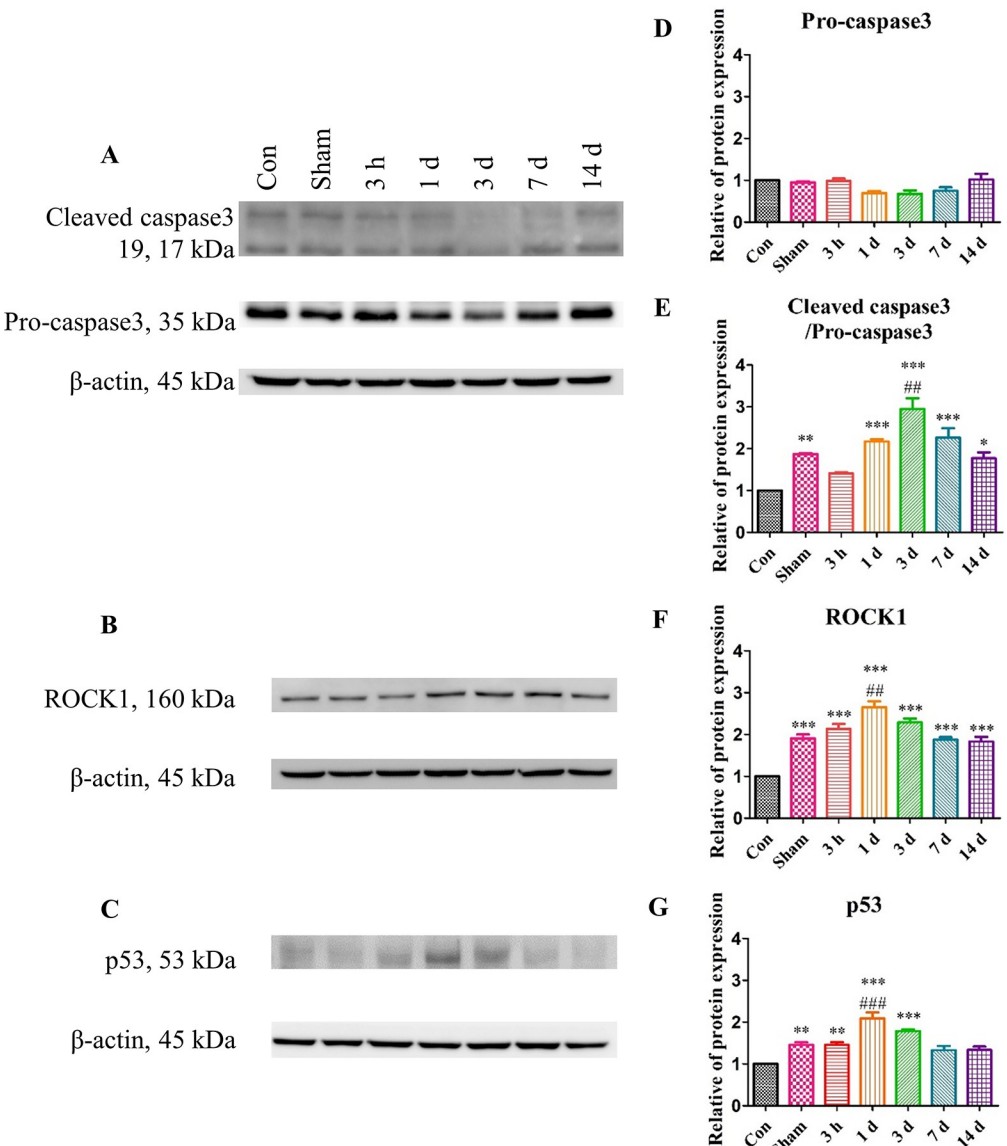

**Fig 5. The downstream factors of the apoptotic signaling pathway were stimulated after TBI induction.** (A-C) The western blotting analyses of cleaved caspase 3, pro-caspase 3, ROCK1, and p53, which are some of the downstream factors of the apoptotic signaling pathway. (D-G) The quantified western blotting analyses. All protein levels were divided by the β-actin level and normalized to that of the Con group. The protein levels increased until Day 3 and decreased by Day 14. Each experiment included three biologically independent mice per group. Significance levels: $*p < 0.05$, $**p < 0.01$, $***p < 0.001$ compared to the Con group; $^{#}p < 0.05$, $^{##}p < 0.01$, $^{###}p < 0.001$ compared to the Sham group.

## The loss and apoptosis of neurons upon TBI induction

To determine the loss and apoptosis of neurons qualitatively, immunohistochemistry was carried out (Figs 6 and 7). To detect the signs of pro-apoptotic signaling, the brain sections were treated with an anti-Bax antibody. Consistent with western blotting analysis, the number of Bax-positive cells was dramatically increased in the TBI-induced brain (Fig 6A and 6C). This indicated that our TBI model indeed promoted apoptosis of neurons. Furthermore, anti-Bcl-2 antibody treatment revealed an increased number of Bcl-2-positive cells in the lesion sites,

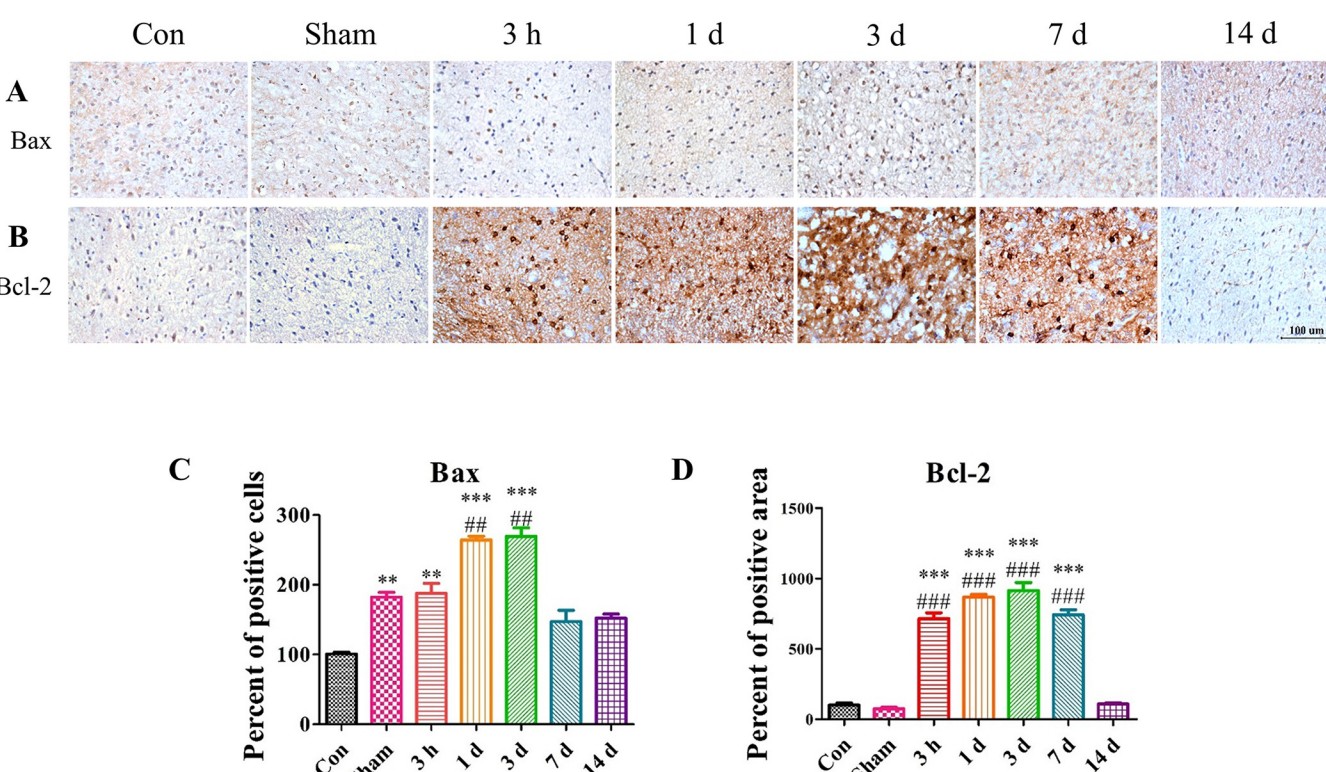

**Fig 6. The expressions of Bax and Bcl-2 were also investigated by immunohistochemistry.** (A and C) The brown dot indicates Bax-positive cells. The Bax expression increased until Day 3 and gradually decreased for the remainder of the study. Each group was normalized to the Con group. (B and D) Bcl-2-positive cells increased until Day 3 and slightly decreased by Day 7. The brown areas and dots indicate the protein-positive areas. Each group was normalized to Con group. Each experiment included three biologically independent mice per group. Significance levels: $^*p < 0.05$, $^{**}p < 0.01$, $^{***}p < 0.001$ compared to the Con group; $^\#p < 0.05$, $^{\#\#}p < 0.01$, $^{\#\#\#}p < 0.001$ compared to the Sham group.

which was not consistent with the western blotting analysis (Fig 6B and 6D). The calculated apoptosis ratio for the TBI group indicated an increase in the numbers of Bax- and Bcl-2-positive cells for up to 3 days post-surgery. While Bax-positive cells showed a reduction after that point, Bcl-2-positive cells were maintained for up to 7 days post-surgery. By14 days post-surgery, the number of Bcl-2-positive cells in the TBI group was comparable to those of the Con and Sham groups.

Furthermore, we investigated the changes in the expression of NeuN, a marker for neuronal nuclei, in the TBI group. It was shown that the morphology of the neurons in the TBI group was irregular compared to the Con and Sham groups, showing a sharp and long shape by 3 hours post-surgery. However, by 1-day post-surgery, these irregular neurons had disappeared and remained absent until 3 days post-surgery. By 7 days post-surgery, the irregular neurons reappeared and had almost completely recovered by 14 days post-surgery (Fig 7A and 7D). Additionally, the expression of MAP2, a microtubule gene, was reduced almost immediately after surgery. Compared to the Con and Sham groups, the TBI group showed a significantly lower number of MAP2-positive cells at earlier time point. However, by 7 days post-surgery, some level of MAP2-positive cells was observed (Fig 7B and 7E), suggesting that the damaged neurons were able to recover. The expression of neurofilament heavy chain (NFH), an actin filament of neurons, was increased up to 1-day post-surgery and remained until 3 days post-surgery. However, by 7 days post-surgery, the level of NFH was reduced and completely recovered by 14 days post-surgery (Fig 7C and 7F).

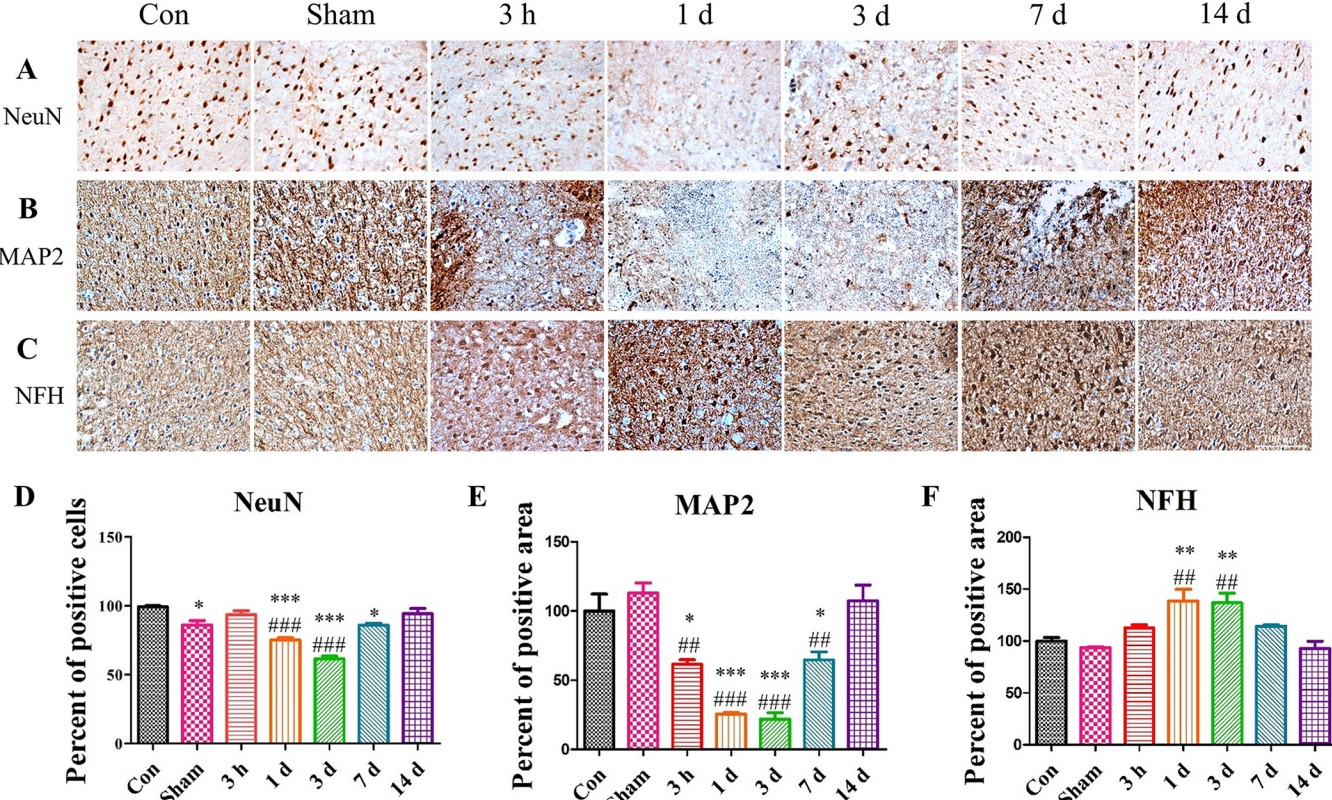

**Fig 7. There was a rapid loss of neurons and disruption in their shape after TBI induction.** (A and D) In the Con and Sham groups, the neuron nuclei showed a spherical shape. In contrast, from 3 hours to 1 day post-surgery, the TBI group's neuron nuclei showed a sharp and long shape. By Day 7, such a shape disappeared in many nuclei. Each group was normalized to the Con group. (B and E) MAP2 expressed in dendrites, somas, and axons were stained and monitored for changes over time. The expression showed abrupt changes every day. Upon TBI induction, MAP2 was not detected initially. However, by Day 14, the protein's expression recovered almost completely. Each group was normalized to Con group. (C and F) The changes in the level of NFH were also monitored over time. As shown in the image, NFH increased dramatically after TBI induction. Each experiment included three biologically independent mice per group. Significance levels: $^*p < 0.05$, $^{**}p < 0.01$, $^{***}p < 0.001$ compared to the Con group; $^{\#}p < 0.05$, $^{\#\#}p < 0.01$, $^{\#\#\#}p < 0.001$ compared to the Sham group.

## Astrocyte and microglia were activated upon completion of apoptosis signaling

Upon TBI induction, we investigated the activity of astrocytes and microglia. The brain sections were stained with antibodies against GFAP and Iba1, which are markers for astrocytes and microglia, respectively (Fig 8). GFAP is an intermediate filament of astrocyte. Astrocyte are considered important in brain diseases [49]. Iba1 is specifically expressed in microglia. Microglia are resident immune cells in the brain and become activated upon brain injury to maintain homeostasis [50]. However, it is accepted that a chronic activation of microglia is detrimental to brain injuries. A shown in Fig 8, the TBI group did not show activated astrocyte and microglia for up to 1-day post-surgery. However, by 3 days post-surgery, we observed that these cells were activated surrounding the lesion sites. Astrocytes and microglia play important roles in brain injuries as the recovery from the injuries is dependent on their phenotypes. These results suggested that astrocytes and microglia were activated upon completion of apoptosis signaling.

## Discussion

In this study, we established a CCI-based TBI mouse model and investigated the changes in motor functions, neurons, astrocyte, and microglia at different time point after the surgery.

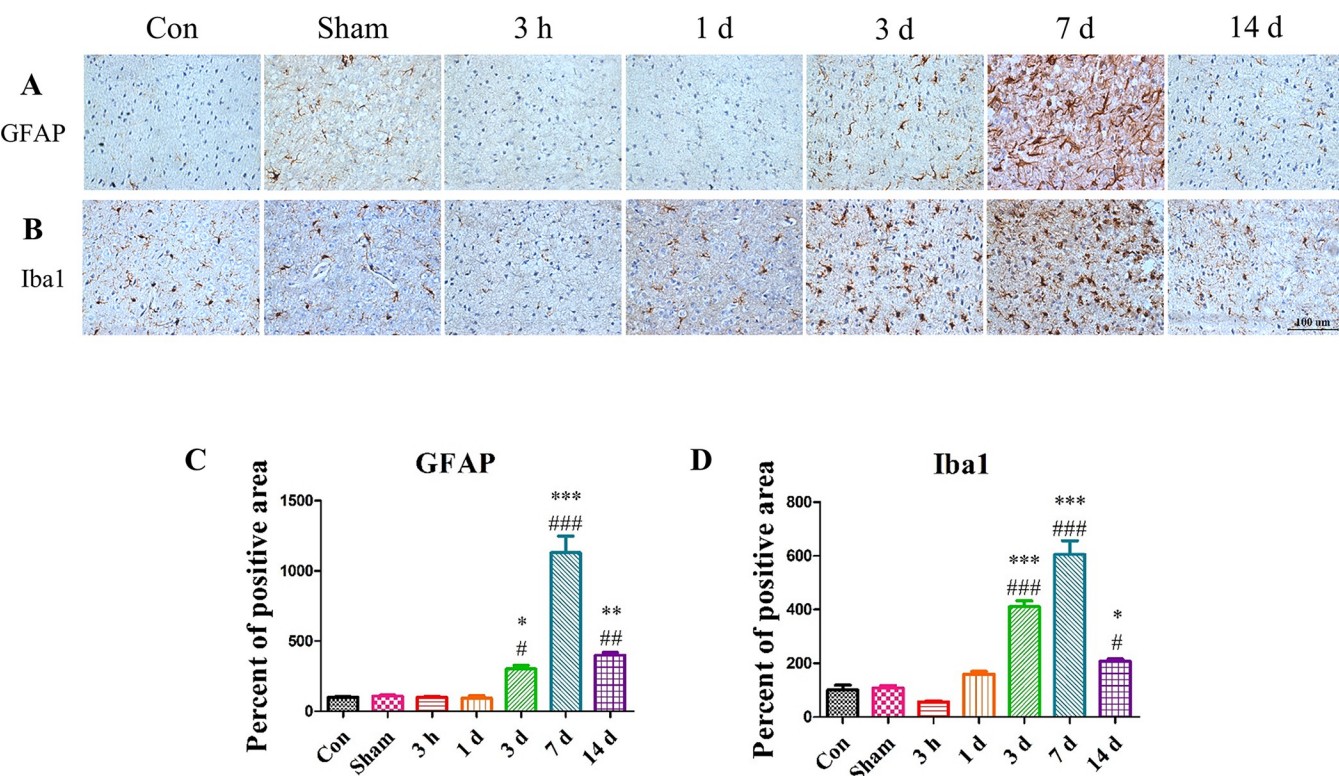

**Fig 8. The changes in the levels of astrocyte and microglia markers after TBI induction.** After the apoptosis signaling was completed, astrocytes were activated. (A and C) GFAP (astrocyte marker) expression was increased on Day 7. (B and D) Iba1 (microglia marker) expression was gradually increased until Day 7 and decreased. Each experiment included three biologically independent mice per group. Significance levels: *$p < 0.05$, **$p < 0.01$, ***$p < 0.001$ compared to the Con group; #$p < 0.05$, ##$p < 0.01$, ###$p < 0.001$ compared to the Sham group.

The TBI group immediately lost left motor functions after the subjects' right motor cortices were damaged. Furthermore, the rotarod test revealed changes in the behaviors of the subjects. Their latency to fall time, which was around 200 seconds before the surgery, decreased to less than 100 seconds after surgery. Also, the subjects showed asymmetric action in the elevated body swing and cylinder test after the surgery. The TBI group showed similar numbers of left and right contacts before the surgery. After the surgery, we observed a marked decline in the number of left contacts, and this symptom persisted throughout the study. In contrast to the partially recovered motor dysfunction, as assessed by the rotarod test, the asymmetric behavior appeared to worsen over time.

Following the primary injury, secondary injuries were first observed at 3 hours post-surgery, and their severity increased by 1-day post-surgery. As shown in Fig 3, the degree of hemorrhaging increased by 1-day post-surgery but appeared to decrease by 3 days post-surgery. By 7 days post-surgery, hemorrhaging had nearly disappeared, leaving only a minor scar by 14 days post-surgery. The infarction regions also gradually decreased over time. TTC staining revealed that the degree of infarction reached its maximum by 1-day post-surgery. The results of this study indicated that the motor dysfunction could not be fully recovered even after the resolution of secondary injuries.

At the protein level, it was confirmed that our TBI model induced activation of the apoptosis signaling pathway. We observed a gradual increase in the Bax/Bcl-2 ratio and Mcl-1 expression, indicating apoptosis of both neurons and microglia (Fig 9). Immunohistochemistry results reflected similar trends as observed in protein expression. The TBI group showed an

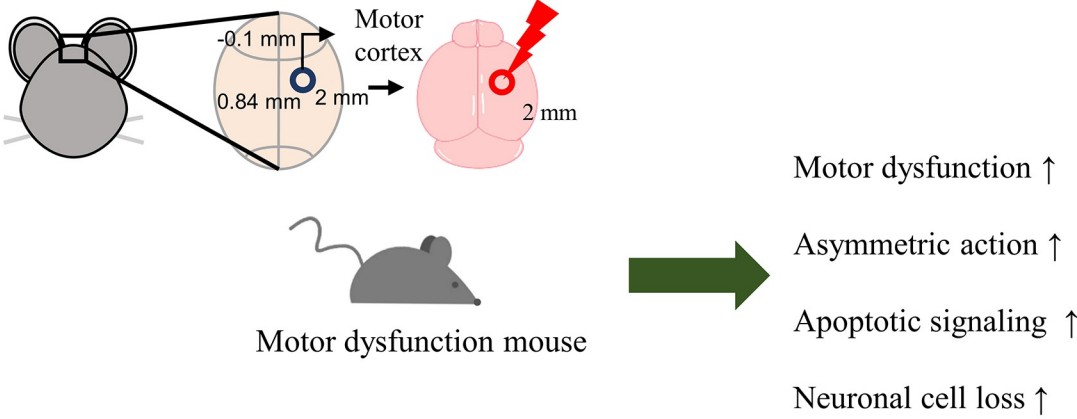

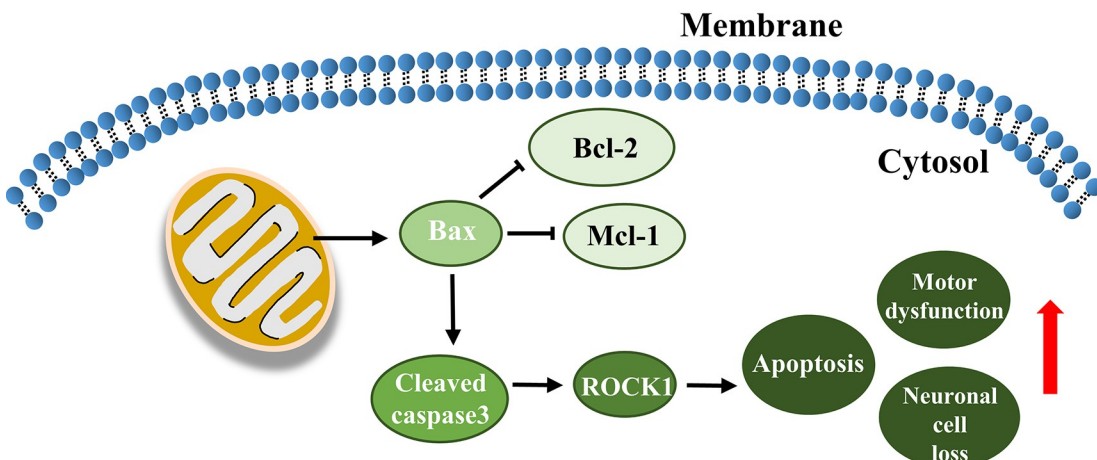

**Fig 9. Experimental set-ups for the animal models of TBI and the results of the cellular levels.** We constructed an impact-acceleration model due to free falling weight. The animals showed a motor dysfunction in the behavioral tests. In addition, the levels of apoptotic signaling-related protein were highly expressed and neuronal cell loss was increased after TBI.

irregular morphology of neurons compared to the Con and Sham groups. While NeuN-positive cells decreased, Bax-positive cells increased by 3 days post-surgery. By 7 days post-surgery, NeuN-positive cells increased, while Bax-positive cells decreased. In case of Bcl-2, contrary to the western blotting analysis, it exhibited a pattern similar to Bax expression. Following TBI surgery, Bcl-2 expression at the lesion sites increased immediately, which was consistent with previous findings [51, 52]. However, the discrepancy in our western blotting analysis may be attributed to not exclusively using the cortex region for analysis. MAP2 shown a similar pattern to NeuN, while NFH exhibited distinct results. This finding suggests the role of NFH as a biomarker for TBI diagnosis [53, 54]. Also, in the case of human TBI, they showed already expression of NFH level [11, 53, 54]. It demonstrates our modeling made by mouse is suitable for researching the care of TBI patients.

Following TBI, there are various changes in cytokines, neurotransmitters, biochemical mediators, and genetics, which contribute to tissue injury through molecular mechanisms [55]. Regarding immune response, damaged-associated molecular patterns (DAMPs)-promoted resident cells rapidly secret cytokines and chemokines upon injury. Microglia and

astrocytes become activated 3–5 days post-injury to defend and repair damaged tissues [56]. At this stage, microglia and astrocytes both show two types of phenotypes: M1 and M2 for microglia, and A1 and A2 for astrocytes. M1 and A1 are pro-inflammatory phenotypes, while M2 and A2 p are anti-inflammatory [57–60].

Pro-inflammatory phenotypes in microglia are often associated with the M1 phenotype. The M1 microglia express CD16 and CD32 and produce inflammatory cytokines, such as interleukin-6 (IL-6), tumor necrosis factor-α (TNF-α), inducible nitric oxide synthase (iNOS), and interleukin-1β (IL-1β). These molecules contribute to inflammation, immune response activation, and tissue damage. In certain contexts, excessive or prolonged inflammation can lead to apoptosis of various cell types, including microglia themselves. Anti-inflammatory phenotypes in microglia are typically associated with the M2 phenotype. The M2 microglia express CD206 (also known as mannose receptor) and Arginase1 and secrete anti-inflammatory cytokines, such as interleukin-4 (IL-4) and interleukin-10 (IL-10). M2 microglia are involved in tissue repair, mitigating inflammation, and promoting the resolution of immune responses. Recently, IL-6 and IL-10, have been identified as potential biomarkers for TBI [61].

Collectively, our study demonstrated motor dysfunction and apoptosis signaling upon TBI induction. While motor deficits showed partial recovery over the time, asymmetric behaviors persisted throughout the study. To facilitate recovery from motor dysfunction, we predict that treatment before 1-day post-TBI will be important. Importantly, the increases in NFH and Bcl-2 expressions observed in this study were analogous to human cases. Thus, we anticipate that our study will be useful for finding an effective treatment option for TBI.

## Conclusion

This study established a CCI-based TBI mouse model to investigate changes in motor functions, neurons, astrocytes, and microglia post-surgery. The TBI group showed immediate motor function loss and persistent asymmetric behavior, with secondary injuries peaking at 1-day post-surgery and resolving by 14 days. Protein analysis revealed apoptosis signaling activation, with increased Bax/Bcl-2 ratio and Mcl-1 expression. NeuN-positive cells decreased initially but increased by 7 days post-surgery, while Bax-positive cells showed the opposite trend. MAP2 is immediately loss but it is recovered soon like NeuN. Previous studies demonstrated that NFH proteins indicated that after TBI it is increased explosively. Microglia and astrocytes exhibited their activation is started after apoptosis signaling finished. The study highlights the importance of early treatment post-TBI and suggests that the mouse model effectively mimics human TBI, aiding in the search for effective treatments.

## Supporting information

**S1 Fig. The whole membrane image of Bcl-2 family.** Related with Fig 4. (A) The Bax and the β-actin of Bax. (B) The Bcl-2 whole membrane and their β-actin. (C) The Mcl-1 and the their β-actin. All data was divided with β-actin and normalized with control group.
(DOCX)

**S2 Fig. The whole membrane image of downstream factor of apoptosis signaling.** Related with Fig 5. (A) The cleaved caspase3, caspase3 and the β-actin of them. (B) The ROCK1 whole membrane and their β-actin. The β-actin was expressed same sample of ROCK1. (C) The p53 and the their β-actin. All data was divided with β-actin and normalized with control group.
(DOCX)

## Author Contributions

**Conceptualization:** Dohee Kim, Han-Seong Jeong, Sujeong Jang.

**Data curation:** Dohee Kim, Jinsu Hwang, Jin Yoo, Jiyun Choi, Mahesh Ramalingam.

**Formal analysis:** Dohee Kim, Jinsu Hwang, Jin Yoo, Jiyun Choi.

**Funding acquisition:** Han-Seong Jeong, Sujeong Jang.

**Investigation:** Han-Seong Jeong, Sujeong Jang.

**Methodology:** Dohee Kim, Jinsu Hwang, Jin Yoo, Jiyun Choi, Mahesh Ramalingam, Hyong-Ho Cho.

**Project administration:** Sujeong Jang.

**Resources:** Hyong-Ho Cho, Byeong C. Kim.

**Software:** Jiyun Choi, Mahesh Ramalingam, Seongryul Kim.

**Validation:** Seongryul Kim, Byeong C. Kim.

**Visualization:** Mahesh Ramalingam, Seongryul Kim.

**Writing – original draft:** Hyong-Ho Cho, Byeong C. Kim, Han-Seong Jeong, Sujeong Jang.

**Writing – review & editing:** Han-Seong Jeong, Sujeong Jang.

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
