## [Decision Letter · Decision Letter 0]

3 Jun 2024

PONE-D-24-14047The Time-Dependent Changes in a Mouse Model of Mild Traumatic Brain Injury with Motor DysfunctionPLOS ONE

Dear Dr. Jang,

Thank you for submitting your manuscript to PLOS ONE. After careful consideration, we feel that it has merit but does not fully meet PLOS ONE’s publication criteria as it currently stands. Therefore, we invite you to submit a revised version of the manuscript that addresses the points raised during the review process.

We look forward to receiving your revised manuscript.

Kind regards,

Firas H Kobeissy, PhD

Academic Editor

PLOS ONE

“This research was supported by grants from the National Research Foundation of Korea (Grant Numbers NRF-2021R1I1A3060435 and NRF-2020R1F1A1076616); a Grant from the Chonnam National University Hospital Biomedical Research Institute (BCRI23041); a Grant from the Korea Institute for Advancement of Technology (KIAT, grant number P0020818) funded by the Korean Government (MOTIE); and a Grant from the Jeollanam-do Science and Technology R&D Project (Development of stem cell-derived new drug) funded by the Jeollanam-do, Korea.”

Reviewers' comments:

Reviewer's Responses to Questions

**Comments to the Author**

1. Is the manuscript technically sound, and do the data support the conclusions?

Reviewer #1: Yes

Reviewer #2: Yes

2. Has the statistical analysis been performed appropriately and rigorously? 

Reviewer #1: Yes

Reviewer #2: Yes

3. Have the authors made all data underlying the findings in their manuscript fully available?

Reviewer #1: Yes

Reviewer #2: Yes

4. Is the manuscript presented in an intelligible fashion and written in standard English?

Reviewer #1: Yes

Reviewer #2: Yes

5. Review Comments to the Author

Reviewer #1: Title: The Time-Dependent Changes in a Mouse Model of Mild Traumatic Brain Injury with Motor Dysfunction

In this manuscript, the authors investigated behavioral and neuropathological changes following a controlled cortical impact (CCI) injury in mice. The study is well documented and does replicate bodies of literature. However, there are many claims should be re-visited. Below are a few major points outlined to be considered.

Major Critiques:

1. Although claimed as mild TBI injury, there are substantial tissue damage, loss, and behavioral alterations, especially visualized from Fig. 3. Therefore, it should be more suitable to refer to as a TBI model or simply a CCI model.

2. Some original references regarding the CCI model are missing, such as PMID: 7629863 and 1787745.

3. One related point is that while it is claimed as a “more concise and effective” mild TBI model, this does not appear to be much more different than numerous established mouse CCI models embedded in literature. Indeed, the behavioral observations along with the histological findings are not new. These may include locomotor dysfunctions, increases in apoptosis, loss of neuronal markers, etc after injury, which are all well characterized in many other studies. This is a major disadvantage of the study that only replicate but not significantly advance understanding of the disease pathophysiology.

Reviewer #2: the manuscript is well drafted and alot of detais is ecountered but there is minimum omission should be deal with them before puplications.

the photo for histopathology and immunohistochemistry should be increasing resolution and increases the description on them.

the paper is good but need to improve the quality of images

western blotting has double bands indicate non specific binding in MCL2

is there is dimenstion or previous report for stereotaxic injection that you have applied

mice genetics should be demostrated and why you shoose this mouse starin .

6. PLOS authors have the option to publish the peer review history of their article (what does this mean?). If published, this will include your full peer review and any attached files.

Reviewer #1: No

Reviewer #2: No

---

## [Author Response · Author response to Decision Letter 0]

17 Jun 2024

Comments from Reviewer 1.

In this manuscript, the authors investigated behavioral and neuropathological changes following a controlled cortical impact (CCI) injury in mice. The study is well documented and does replicate bodies of literature. However, there are many claims should be re-visited. Below are a few major points outlined to be considered.

I appreciate your detailed review and valuable comments. All the sentences that I changed following your recommendation the instruction of the journal, we have marked as blue color in the manuscript.

Question 1:

Although claimed as mild TBI injury, there are substantial tissue damage, loss, and behavioral alterations, especially visualized from Fig. 3. Therefore, it should be more suitable to refer to as a TBI model or simply a CCI model.

Answer: Thank you for your detailed review and valuable comments. Following your recommendation, we have changed TBI model instead of mild TBI model in all the manuscript. 

Question 2:

Some original references regarding the CCI model are missing, such as PMID: 7629863 and 1787745.

Answer: Thank you for your detailed review and valuable comments. Following your recommendation, we have added the original references in the Introduction and Materials and Methods and ordered all the references.

Question 3:

One related point is that while it is claimed as a “more concise and effective” mild TBI model, this does not appear to be much more different than numerous established mouse CCI models embedded in literature. Indeed, the behavioral observations along with the histological findings are not new. These may include locomotor dysfunctions, increases in apoptosis, loss of neuronal markers, etc after injury, which are all well characterized in many other studies. This is a major disadvantage of the study that only replicate but not significantly advance understanding of the disease pathophysiology.

Answer: Thank you for your detailed review and valuable comments. In this study, we evaluated the mouse model which had an accurate area of damage with 2.0 mm of diameter in the brain. Following the results, the mice showed same motor dysfunction with increased apoptotic signaling pathway. In many studies including our results, there are still remain limitations, however, our model showed very concise and delicate sign and symptom than others. 

Comments from Reviewer 2.

the manuscript is well drafted and alot of detais is ecountered but there is minimum omission should be deal with them before puplications.

I appreciate your detailed review and valuable comments. All the sentences that I changed following your recommendation, we have marked as blue color in the manuscript.

Question 1:

the photo for histopathology and immunohistochemistry should be increasing resolution and increases the description on them.

Answer: Thank you for your detailed review and valuable comments. Following your recommendation, we have increased the resolution and provided more detail description of the figures. 

Question 2:

the paper is good but need to improve the quality of images

Answer: Thank you for your detailed review and valuable comments. Following your recommendation, we have increased the quality of images. We will provide improved quality of images, separately. 

Question 3:

western blotting has double bands indicate non specific binding in MCL2

Answer: Thank you for your detailed review and valuable comments. Our previous studies, the western blotting analysis of MCL2 had showed double bands, so we described and provide the references of the results. 

Question 4:

is there is dimenstion or previous report for stereotaxic injection that you have applied

Answer: Thank you for your detailed review and valuable comments. In this study, we used the stereotaxic system to calculate the exact region of impact not to inject. For the next work, we are working about the therapeutic effect of drugs in the TBI model to use a stereotaxic, which administered by intracerebroventricular injection with 10 ul of Hamilton syringe (701N, Lot # 697293, Hamilton Company, NV, USA) on a stereotaxic syringe pump (KD Scientific Inc., MA, USA). 

Question 5:

mice genetics should be demostrated and why you shoose this mouse starin .

Answer: Thank you for your detailed review and valuable comments. Following your recommendation, we have added the reference and the reason to choose the mouse strain in the Materials and Methods.

---

## [Decision Letter · Decision Letter 1]

9 Jul 2024

The Time-Dependent Changes in a Mouse Model of Traumatic Brain Injury with Motor Dysfunction

PONE-D-24-14047R1

Dear Dr. Jang,

We’re pleased to inform you that your manuscript has been judged scientifically suitable for publication and will be formally accepted for publication once it meets all outstanding technical requirements.

Kind regards,

Firas H Kobeissy, PhD

Academic Editor

PLOS ONE

Additional Editor Comments (optional):

Reviewers' comments:

Reviewer's Responses to Questions

**Comments to the Author**

1. If the authors have adequately addressed your comments raised in a previous round of review and you feel that this manuscript is now acceptable for publication, you may indicate that here to bypass the “Comments to the Author” section, enter your conflict of interest statement in the “Confidential to Editor” section, and submit your "Accept" recommendation.

Reviewer #1: All comments have been addressed

Reviewer #2: All comments have been addressed

2. Is the manuscript technically sound, and do the data support the conclusions?

Reviewer #1: Yes

Reviewer #2: Yes

3. Has the statistical analysis been performed appropriately and rigorously? 

Reviewer #1: Yes

Reviewer #2: I Don't Know

4. Have the authors made all data underlying the findings in their manuscript fully available?

Reviewer #1: Yes

Reviewer #2: Yes

5. Is the manuscript presented in an intelligible fashion and written in standard English?

Reviewer #1: Yes

Reviewer #2: (No Response)

6. Review Comments to the Author

Reviewer #1: The authors have addressed all comments appropriately, with additional new experiments and revision of the text.

Reviewer #2: It is a good paper, the author addressed all the required information. the paper should be puplished to dispaly all this information for the scienttists.

7. PLOS authors have the option to publish the peer review history of their article (what does this mean?). If published, this will include your full peer review and any attached files.

Reviewer #1: **Yes: **Hailong Song

Reviewer #2: **Yes: **Mohamed Fawzy Hamed

---

## [Editor Report · Acceptance letter]

16 Jul 2024

PONE-D-24-14047R1 

PLOS ONE

Dear Dr. Jang, 

I'm pleased to inform you that your manuscript has been deemed suitable for publication in PLOS ONE. Congratulations! Your manuscript is now being handed over to our production team.

Kind regards, 

on behalf of

Dr. Firas H Kobeissy 

Academic Editor

PLOS ONE